**Data Availability Statement:** The data used in this study cannot be made available in the manuscript, the supplemental files, or in a public repository due

# 6-month mortality and readmissions of hospitalized COVID-19 patients: A nationwide cohort study of 8,679 patients in Germany

**Christian Günster**[1◉], **Reinhard Busse**[2◉], **Melissa Spoden**[1], **Tanja Rombey**[2], **Gerhard Schillinger**[1], **Wolfgang Hoffmann**[3], **Steffen Weber-Carstens**[4], **Andreas Schuppert**[5], **Christian Karagiannidis**[6]*

1 Research Institute of the Local Health Care Funds, Federal Association of the Local Health Care Funds, Berlin, Germany, 2 Department of Health Care Management, Technische Universität Berlin, Berlin, Germany, 3 Section Epidemiology of Health Care and Community Health, Institute for Community Medicine, University Medicine Greifswald, Greifswald, Germany, 4 Department of Anesthesiology and Operative Intensive Care Medicine, Charité - Universitätsmedizin Berlin, Berlin, Germany, 5 Institute for Computational Biomedicine II, University Hospital RWTH Aachen University, Aachen, Germany, 6 Department of Pneumology and Critical Care Medicine, Cologne-Merheim Hospital, Kliniken der Stadt Köln, Witten/ Herdecke University Hospital, Cologne, Germany

◉ These authors contributed equally to this work.
* Christian.Karagiannidis@uni-wh.de, karagiannidisc@kliniken-koeln.de

## Abstract

### Background

COVID-19 frequently necessitates in-patient treatment and in-patient mortality is high. Less is known about the long-term outcomes in terms of mortality and readmissions following in-patient treatment.

### Aim

The aim of this paper is to provide a detailed account of hospitalized COVID-19 patients up to 180 days after their initial hospital admission.

### Methods

An observational study with claims data from the German Local Health Care Funds of adult patients hospitalized in Germany between February 1 and April 30, 2020, with PCR-confirmed COVID-19 and a related principal diagnosis, for whom 6-month all-cause mortality and readmission rates for 180 days after admission or until death were available. A multivariable logistic regression model identified independent risk factors for 180-day all-cause mortality in this cohort.

### Results

Of the 8,679 patients with a median age of 72 years, 2,161 (24.9%) died during the index hospitalization. The 30-day all-cause mortality rate was 23.9% (2,073/8,679), the 90-day rate was 27.9% (2,425/8,679), and the 180-day rate, 29.6% (2,566/8,679). The latter was 52.3% (1,472/2,817) for patients aged ≥80 years 23.6% (1,621/6,865) if not ventilated

to German data protection laws (Bundesdatenschutzgesetz). Therefore, they are stored on a secure drive in the Wissenschaftliches Institut der AOK to facilitate replication of the results. Generally, access to data of statutory health insurance funds for research purposes is possible only under the conditions defined in the German Social Law (SGB V § 287). Requests for data access can be sent as a formal proposal, specifying the recipient and purpose of the data transfer, to the appropriate data protection agency. Access to the data used in this study can only be provided to external parties under the conditions of the cooperation contract of this research project and after written approval by the AOK. For assistance in obtaining access to the data, please contact wido@wido.bv.aok.de.

**Funding:** Institutional support and physical resources were provided by the University Witten/ Herdecke and Kliniken der Stadt Köln, the Federal Association of the Local Health Care Funds and the Technical University of Berlin. The latter also received a grant from the Berlin University Alliance (112_PreEP_Corona). Article processing fees were funded by the authors. No funding source had a role in the design or conduct of the study; data collection, management, analysis, or interpretation; or the preparation, review, or approval of the manuscript.

**Competing interests:** RB reports grants from Berlin University Alliance, during the conduct of the study; grants from Federal Ministry of Research and Education, grants from Federal Ministry of Health, grants from Innovation Fonds of the Federal Joint Committee, grants from World Health Organization, outside the submitted work, AS reports grants from Bayer AG, outside the submitted work. CK reports personal fees from Maquet, personal fees from Xenios, personal fees from Bayer, non-financial support from Speaker of the German register of ICUs, grants from German Ministry of Research and Education, during the conduct of the study. CG, MS, TR, GS, WH, and SWC have nothing to disclose.

during index hospitalization, but 53.0% in case of those ventilated invasively (853/1,608). Risk factors for the 180-day all-cause mortality included coagulopathy, BMI $\geq$ 40, and age, while the female sex was a protective factor beyond a fewer prevalence of comorbidities. Of the 6,235 patients discharged alive, 1,668 were readmitted a total of 2,551 times within 180 days, resulting in an overall readmission rate of 26.8%.

## Conclusions

The 180-day follow-up data of hospitalized COVID-19 patients in a nationwide cohort representing almost one-third of the German population show significant long-term, all-cause mortality and readmission rates, especially among patients with coagulopathy, whereas women have a profoundly better and long-lasting clinical outcome compared to men.

## Introduction

### Background

Within one year, the SARS-CoV-2 pandemic has affected more than 125 million people worldwide. During the first wave in spring 2020, hospitalization rates were high, reaching up to 70% in France, 55% in Spain, 50% in the UK and 20% in Germany until the end of April [1]. Mortality rates of hospitalized patients were also high at 12.5% in France [2], and more than 20% in the UK and Germany [3, 4], especially among patients requiring mechanical ventilation with up to 50%. Little is known about the long-term outcome of hospitalized COVID-19 patients, though, in general, there is increasing evidence of a long-COVID syndrome, affecting different organ systems.

Recently, the 6-month follow-up data of the first 1,733 hospitalized COVID-19 patients from Wuhan were published [5]. COVID-19 survivors suffered mainly from fatigue or muscle weakness, sleep difficulties, and anxiety or depression. These data are in line with recent data of an increased risk for neurological and psychiatric disorders after 6 months [6–10]. In case of severely ill COVID-19 patients, it has been found among 167 COVID-19 patients admitted to an intensive care unit (ICU) that older age and thrombocytopenia, among others, were significant risk factors for mortality at 28, 90 and 180 days [11]. Furthermore, in a multicentre-study from five European countries of hospitalized COVID-19 patients requiring extracorporeal membrane oxygenation (ECMO) showed that advanced age and low pH before ECMO were also associated with an increased risk of mortality within 6 months, with an overall rate of 53% (70/132) [12].

For large cohorts outside Wuhan, follow-up data have been reported up to a maximum of three months. Ninety-day *post-admission* outcomes have been reported from various European countries, albeit often for single hospitals. The 90-day mortality rate ranged from 11% in Spain [13] to 29% in Denmark [14] (both single-centre studies) for all hospitalized patients, and was 27% in Sweden [15], 31% in Belgium, France and Switzerland (both multi-centre studies) [16], and 35% in a Danish single-centre study for patients treated in the ICU [14]. An analysis of more than 100,000 hospitalized COVID-19 patients in the United States revealed a readmission rate of 9% to the same hospital 6 months post-discharge [17].

### Objectives

Since a large cohort of 6-month follow-up data is currently lacking, the aim of this observational study was to determine the 6-month all-cause mortality and readmission rates of

hospitalized COVID-19 patients who completed hospital treatment following a confirmed COVID-19 diagnosis, focusing particularly on patients requiring mechanical ventilation. Furthermore, factors associated with 6-month all-cause mortality were evaluated.

## Materials and methods

### Study design

This was a retrospective observational cohort study, using claims data. The study was investigator-initiated without any funding. It was approved by the Ethics Committee of the Witten/ Herdecke University (research ethics board number 92/2020).

### Setting

We used nationwide anonymous administrative claims data for in-patient episodes (including diagnoses, procedures, length of stay, transfers and discharge type), and core data (including age, gender, insurance status and survival status) of the sickness funds group "German local healthcare funds" (Allgemeine Ortskrankenkassen, AOK). In Germany, health insurance is obligatory and nearly every inhabitant has health insurance [18].

AOK is the largest sickness funds group within Germany's statutory health insurance system. It provides statutory health insurance for roughly 32 percent of the German population, for whom it is representative when considering the factor age [19, 20]. Furthermore, membership in a sickness fund is open to anyone regardless of factors such as professional affiliation, income, or comorbidities. Sickness funds are obliged to accept any applicant and charge the same basic contribution rate [18]. According to the German accounting method for the health care system, all diagnoses, outcomes, and procedures must be reported to the sickness funds, as required by law. Strict legal requirements and verification by both hospitals and sickness funds aim to minimize any bias introduced by false coding and other systemic errors relating to the data. In addition, we checked data plausibility and consistency before analyzing them. Diagnoses were coded according to the 10th revision of the International Classification of Diseases (ICD-10-GM) and procedures according to the International Classification of Procedures in Medicine, the "Operationen- und Prozedurenschlüssel" (OPS).

### Study population

For the analyses, we included only patients with COVID-19 infection as confirmed by reverse transcription polymerase chain reaction (PCR) tests (diagnosis code U07.1), who were at least 18 years old, and were admitted to hospital for treatment of COVID-19 between February 1, 2020 and April 30, 2020, both dates included, and were discharged by June 30, 2020.

As COVID-19 cannot be coded as a principal diagnosis, we defined hospitalization for COVID-19 as an admission with a COVID-19-related principal diagnosis of respiratory failure, pulmonary embolism, viral infection, sepsis or renal failure during the initial ("index") hospitalization with confirmed COVID-19 infection. A list of all included diagnoses is provided in S1 Table. The selection of patients was done to include only patients in whom COVID-19 was the primary reason for their hospital stay and to exclude those in whom COVID-19 was an incidental finding likely to be unrelated to their hospital stay. The unit of analysis was the individual patient. Since one person might have had several hospital stays during the observation period due to a transfer from one hospital to another, we grouped adjacent completed hospital stays into one patient. Each patient was followed up for 180 days after admission for survival status, and after discharge for readmissions. Patients had to be

continuously insured with AOK for that period of time, unless they died earlier while still insured with AOK. This was a complete survey of all AOK insured.

## Endpoints

All-cause mortality was measured as in-hospital mortality, i.e. death occurring during the index hospitalization, as well as in or out-of-hospital mortality within 30 days, 90 days and 180 days after the day of initial admission. The date of out-of-hospital death was extracted from the core data of the sickness funds.

For readmissions, we followed patients discharged alive from the index hospitalization for 180 days after the day of discharge. The discharge date was chosen because the patients' length of index hospital stay varied widely. We determined both how many patients were readmitted at least once, and how many readmissions occurred in total for any particular cause, and for readmission with ventilation or potentially COVID-19-related systemic, respiratory, renal and neurological, gastrointestinal and liver as well as cardiovascular diagnoses only (S1 Table).

## Statistical analysis

Baseline characteristics are described in terms of means and standard deviations (SD), and medians and inter-quartile ranges (IQR) for continuous variables, and in terms of proportions for categorical variables. Characteristics are shown for the whole study population and for patients with and without mechanical ventilation. For ventilated patients, two subgroups were formed: (a) patients with non-invasive mechanical ventilation only (NIV), and (b) patients with invasive mechanical ventilation (IMV), including those with non-invasive mechanical ventilation failure. Baseline characteristics are defined on the basis of conditions and interventions during index hospitalization and include gender, age groups (18–59 years, 60–69 years, 70–79 year, $\geq$80 years), selected Elixhauser comorbidities [21, 22], and type of ventilation. In addition, we report on other procedures, such as dialysis or ECMO, as well as further comorbidities and complications, such as septic shock, acute respiratory distress syndrome (ARDS), or renal failure post medical procedures. See S1 Table.

We report all-cause mortality (in-hospital, 30/90/180 days after admission) for patients grouped according to their baseline characteristics.

Kaplan-Meier curves for survival up to 180 days after the initial admission are presented by gender, age, type of ventilation, and selected Elixhauser comorbidities.

Owing to non-parallel Kaplan-Meier curves and a significant deviation from the proportional hazards assumption, no Cox-proportional hazard model could be estimated. The proportional hazards assumption was evaluated visually on plots of log (-log[survival]) versus log of survival time adjusted for covariates and by a global test based on Schoenfeld residuals. Instead, multivariable logistic regression was used to model the odds of the binary endpoint of 180-day all-cause mortality after admission as a function of age, sex, body mass index (BMI) categories ($30\leq34$, $35\leq39$, $\geq40$ kg/m2) and Elixhauser comorbidities present at index hospitalization. We used cluster-robust standard errors to account for the clustering of patients in hospitals. Comorbidity conditions were defined as binary variables. A model including all potential risk factors was estimated first, with subsequent removal of risk factors that did not prove to be statistically significant ($p \geq 0.05$). A Wald test was performed to confirm that the removal of these risk factors from the model did not result in any substantial loss in model fit. Adjusted odds ratios (OR) and 95% confidence intervals (CIs) were calculated and summarized in a forest plot. To evaluate the performance of the model, the area under the curve (AUC) was used as a measure of discrimination and the squared Pearson correlation ($R^2$) between 180-day all-cause mortality and the log-odds of predicted mortality was used as a

measure of the explained variation. All analyses were performed using STATA 16.0 (Stata-Corp, College Station, Texas).

## Results

### Study cohort

The patient flow is displayed in Fig 1. Out of the 1.075 million AOK-insured persons with at least one in-patient admission between February 1 and April 30, 2020, we identified 11,459 adult patients with PCR-confirmed COVID-19 diagnosis who were discharged before June 30, 2020. We excluded 2,572 patients who did not suffer from a COVID-19-related principal diagnosis but were admitted for diagnoses such as congestive heart disease (n = 161) or femur fracture (n = 139). Furthermore, we had to exclude 208 patients, who could not be followed for the full 6-month period after their initial admission or until death. In total, 8,679 patients were confirmed eligible and included in the analysis.

### Demographics and comorbidity

The patients' demographic characteristics are shown in Table 1. The cohort comprised slightly more men (4,641/8,679; 53.5%) than women (4,038/8,679; 46.5%). The median age was 72 years (IQR 57 to 82), the largest share of patients being in the age group of 80 years and older (2,817/8,679; 32.5%). The most observed comorbidities were hypertension (4,920/8,679;

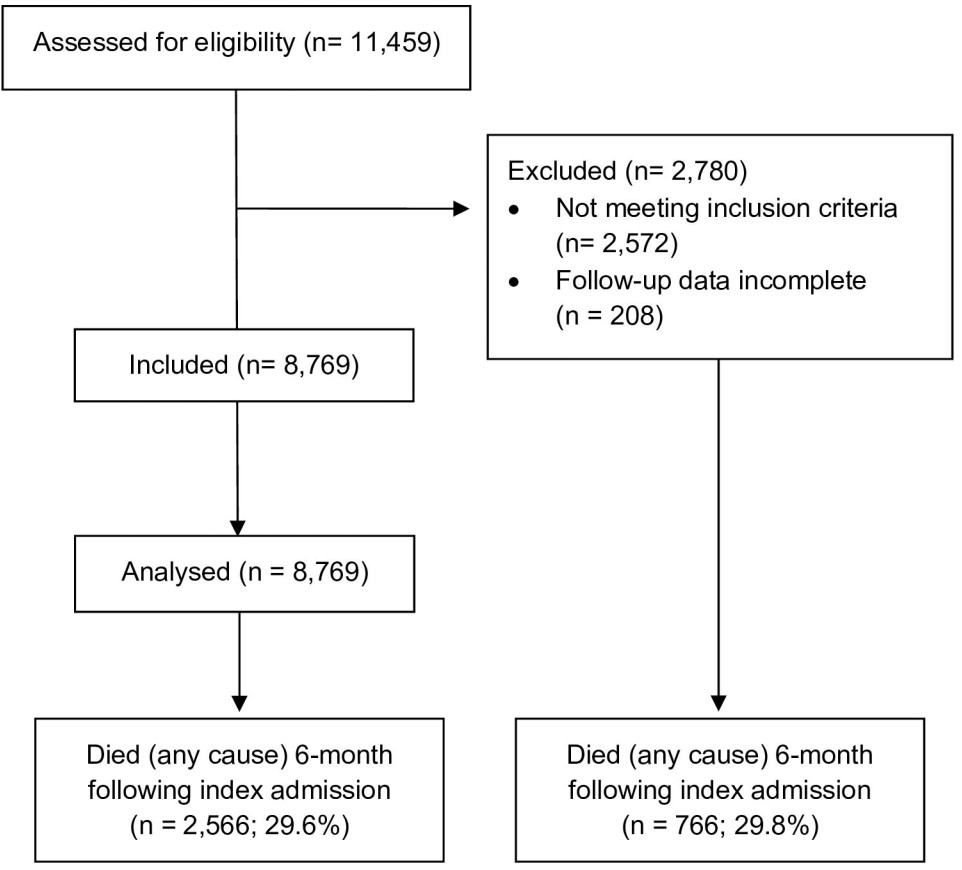

**Fig 1. Patient flow diagram.**

56.7%), fluid and electrolyte disorders (4,641/8,679; 53.5%), diabetes mellitus (uncomplicated: 1,912/8,679; 22.0%; complicated: 734/8,679; 8.5%), cardiac arrhythmia (2,373/8,679; 27.3%), renal failure (1,994/8,679; 23.0%), and congestive heart failure (1,652/8,679; 19.0%).

## Index hospitalization

Median length of stay of the index hospitalization was 10 days (IQR 6 to 20); the mean length of stay was 16.5 (SD 19.4) days (Table 1). 6,865 (79.1%) patients were treated without mechanical ventilation and 1,814 (20.9%) were treated with mechanical ventilation, of whom 1,608 (18.5%) received IMV and 206 (2.4%) received NIV only. The patients' demographic characteristics stratified by ventilation status can be found in S2 Table. Dialysis was performed in 9.7% (838/8,679) and ECMO in 1.8% (160/8,679). Frequent complications during the index hospitalization included septic shock (1,406/8,679; 16.2%), ARDS (1,329/8,679; 15.3%), and acute renal failure (1,252/8,679; 14.4%).

## All-cause mortality

Of the 8,679 included patients, 2,161(24.9%) died during the index hospitalization (Table 1). Measured from the day of initial admission, 30-day all-cause mortality was 23.9% (2,073/8,679), 90-day all-cause mortality was 27.9% (2,425/8,679), and 180-day all-cause mortality was 29.6% (2,566/8,679).

Women's survival rates were about 5 percentage points higher than men's at all three time points (Fig 2A). Survival was also strongly associated with age. Patients in the age group of 18–59 years had a 180-day all-cause mortality of 6.7% (1,65/2,451), which increased to 19.3% (291/1,506) in patients aged 60–69 years, 33.5% (638/1,905) in patients aged 70–79, and 52.3% (1,472/2,817) in patients aged 80 years or above (Fig 2B). The relative difference between in-hospital and 180-day all-cause mortality was also the highest in this age group with an additional 9.7% of patients dying during follow-up (Table 1). Patients who were not ventilated had a better survival record than ventilated patients (Fig 2C), their 180-day all-cause mortality being 23.6% (1,621/6,865) compared to 52.1% (945/1,814) of ventilated patients. Amongst ventilated patients, 30-, 90- and 180-day all-cause mortality rates were about 8% percentage points lower in patients treated with NIV compared to IMV.

Among the comorbidities present in at least 5% of the sample, the greatest difference between 30-day and 180-day all-cause mortality was observed for the following comorbidities: Coagulopathy, congestive heart failure, other neurological disorders, renal failure, and complicated diabetes mellitus (Table 1). Patients with coagulopathy had the largest increase in 180-day all-cause mortality, with an additional 15.6% (97/623) of patients dying between 30 and 180 days (Fig 2D). Survival was considerably better for patients without these comorbidities. For patients with coagulopathy, 180-day all-cause mortality was 49.3% (307/623; Fig 2D), with congestive heart failure 49.8% (823/1,652; Fig 2E), with other neurological disorders 46.5% (312/671; Fig 2F), with renal failure 47.2% (942/1,994; Fig 2G), and with complicated diabetes 45.4% (333/734; Fig 2H).

## All-cause mortality among ventilated patients

In a subgroup analysis of ventilated patients only, the observed trends regarding gender, age, and ventilation type (non-invasive/invasive) persisted (Fig 3A–3C), while the differences narrowed (congestive heart failure and renal failure; Fig 3E and 3G) or vanished in the analyses related to comorbidity (Fig 3D, 3F and 3H). For patients with coagulopathy, the 30-day all-cause mortality was even lower than that of patients with no coagulopathy (Fig 3D); for patients with other neurological disorders, this was even observed beyond Day 90 (Fig 3F).

**Table 1. Patient characteristics and all-cause mortality.**

| Patients hospitalized for a COVID-19-related principal diagnosis | Total | In-hospital all-cause mortality | 30-day all-cause mortality after hospital admission | 90-day all-cause mortality after hospital admission | 180-day all-cause mortality after hospital admission |
|---|---|---|---|---|---|
| | (n = 8,679) | (n = 2,161) | (n = 2,073) | (n = 2,425) | (n = 2,566) |
| | | N (row percentage of total patients) | | | |
| Total (N) | 8,679 (100.0%) | 2,161 (24.9%) | 2,073 (23.9%) | 2,425 (27.9%) | 2,566 (29.6%) |
| Male | 4,641 (53.5%) | 1,293 (27.9%) | 1,209 (26.1%) | 1,416 (30.5%) | 1,489 (32.1%) |
| Female | 4,038 (46.5%) | 868 (21.5%) | 864 (21.4%) | 1,009 (25.0%) | 1,077 (26.8%) |
| Age, years | | (value refers to column total) | | | |
| Mean (SD) | 68.6 (16.6) | 78.4 (11.0) | 79.2 (10.6) | 78.8 (10.9) | 78.9 (10.9) |
| Median (IQR) | 72.0 (57.0–82.0) | 81.0 (73.0–86.0) | 81.0 (75.0–86.0) | 81.0 (73.0–86.0) | 81.0 (73.0–86.0) |
| Age groups, years | | N (row percentage of total patients) | | | |
| 18–59 years | 2,451 (28.2%) | 152 (6.2%) | 123 (5.0%) | 158 (6.4%) | 165 (6.7%) |
| 60–69 years | 1,506 (17.4%) | 263 (17.5%) | 221 (14.7%) | 282 (18.7%) | 291 (19.3%) |
| 70–79 years | 1,905 (21.9%) | 548 (28.8%) | 504 (26.5%) | 598 (31.4%) | 638 (33.5%) |
| 80 years and older | 2,817 (32.5%) | 1,198 (42.5%) | 1,225 (43.5%) | 1,387 (49.2%) | 1,472 (52.3%) |
| Elixhauser comorbidities | | N (row percentage of total patients) | | | |
| Hypertension | 4,920 (56.7%) | 1,296 (26.3%) | 1,227 (24.9%) | 1,478 (30.0%) | 1,577 (32.1%) |
| Fluid and electrolyte disorders | 4,641 (53.5%) | 1,445 (31.1%) | 1,343 (28.9%) | 1,618 (34.9%) | 1,710 (36.8%) |
| Cardiac arrhythmias | 2,373 (27.3%) | 940 (39.6%) | 876 (36.9%) | 1,035 (43.6%) | 1,101 (46.4%) |
| Renal failure | 1,994 (23.0%) | 770 (38.6%) | 746 (37.4%) | 878 (44.0%) | 942 (47.2%) |
| Diabetes, uncomplicated | 1,912 (22.0%) | 575 (30.1%) | 537 (28.1%) | 631 (33.0%) | 664 (34.7%) |
| Congestive heart failure | 1,652 (19.0%) | 700 (42.4%) | 631 (38.2%) | 769 (46.5%) | 823 (49.8%) |
| Chronic pulmonary disease | 1,188 (13.7%) | 360 (30.3%) | 327 (27.5%) | 389 (32.7%) | 420 (35.4%) |
| Diabetes, complicated | 734 (8.5%) | 280 (38.1%) | 261 (35.6%) | 312 (42.5%) | 333 (45.4%) |
| Other neurological disorders | 671 (7.7%) | 255 (38%) | 235 (35.0%) | 296 (44.1%) | 312 (46.5%) |
| Coagulopathy | 623 (7.2%) | 285 (45.7%) | 210 (33.7%) | 290 (46.5%) | 307 (49.3%) |
| Depression | 601 (6.9%) | 104 (17.3%) | 101 (16.8%) | 125 (20.8%) | 144 (24.0%) |
| Liver disease | 411 (4.7%) | 184 (44.8%) | 139 (33.8%) | 187 (45.5%) | 194 (47.2%) |
| Pulmonary circulation disorders | 358 (4.1%) | 157 (43.9%) | 126 (35.2%) | 164 (45.8%) | 175 (48.9%) |
| Weight loss | 351 (4.0%) | 116 (33%) | 98 (27.9%) | 125 (35.6%) | 142 (40.5%) |
| Metastatic cancer | 53 (0.6%) | 26 (49.1%) | 23 (43.4%) | 32 (60.4%) | 37 (69.8%) |
| Further comorbidities | | N (row percentage of total patients) | | | |
| Cognitive impairment | 2,207 (25.4%) | 718 (32.5%) | 719 (32.6%) | 874 (39.6%) | 947 (42.9%) |
| Delir, anoxia encephalopathy, somnolence, sopor and coma | 1,058 (12.2%) | 379 (35.8%) | 330 (31.2%) | 429 (40.5%) | 459 (43.4%) |
| BMI ≥ 40 | 173 (2.0%) | 59 (34.1%) | 45 (26.0%) | 60 (34.7%) | 62 (35.8%) |
| Length of stay of index hospitalization | | (value refers to column total) | | | |
| Mean (SD) | 16.5 (19.5) | 12.8 (14.2) | 9.6 (7.3) | 13.0 (12.7) | 14.0 (15.1) |
| Median (IQR) | 10.0 (6.0–20.0) | 8.0 (4.0–16.0) | 7.0 (4.0–14.0) | 9.0 (4.0–17.0) | 9.0 (5.0–18.0) |
| Ventilation during index hospitalization | | N (row percentage of total patients) | | | |
| Not ventilated | 6,865 (79.1%) | 1,248 (18.2%) | 1,304 (19.0%) | 1,503 (21.9%) | 1,621 (23.6%) |
| Invasive ventilation | 1,608 (18.5%) | 832 (51.7%) | 695 (43.2%) | 836 (52.0%) | 853 (53.0%) |
| Only non-invasive ventilation | 206 (2.4%) | 81 (39.3%) | 74 (35.9%) | 86 (41.7%) | 92 (44.7%) |
| Procedures during index hospitalization | | N (row percentage of total patients) | | | |
| Dialysis | 838 (9.7%) | 474 (56.6%) | 382 (45.6%) | 482 (57.5%) | 508 (60.6%) |
| Tracheostomy | 582 (6.7%) | 221 (38.0%) | 118 (20.3%) | 217 (37.3%) | 230 (39.5%) |

*(Continued)*

**Table 1.** (Continued)

| Patients hospitalized for a COVID-19-related principal diagnosis | Total | In-hospital all-cause mortality | 30-day all-cause mortality after hospital admission | 90-day all-cause mortality after hospital admission | 180-day all-cause mortality after hospital admission |
|---|---|---|---|---|---|
| | (n = 8,679) | (n = 2,161) | (n = 2,073) | (n = 2,425) | (n = 2,566) |
| Extracorporeal membrane oxygenation | 160 (1.8%) | 102 (63.8%) | 68 (42.5%) | 100 (62.5%) | 102 (63.8%) |
| Haemofiltration | 51 (0.6%) | 37 (72.5%) | 33 (64.7%) | 37 (72.5%) | 37 (72.5%) |
| Complications during index hospitalization | | N (row percentage of total patients) | | | |
| Septic shock | 1,406 (16.2%) | 746 (53.1%) | 635 (45.2%) | 764 (54.3%) | 787 (56.0%) |
| ARDS | 1,329 (15.3%) | 692 (52.1%) | 590 (44.4%) | 696 (52.4%) | 708 (53.3%) |
| Renal failure post procedure | 1,252 (14.4%) | 739 (59.0%) | 637 (50.9%) | 766 (61.2%) | 795 (63.5%) |
| Lung embolism | 188 (2.2%) | 74 (39.4%) | 61 (32.4%) | 78 (41.5%) | 81 (43.1%) |
| Intracerebral bleeding, cerebral infarction, stroke | 122 (1.4%) | 60 (49.2%) | 44 (36.1%) | 63 (51.6%) | 65 (53.3%) |
| Acute myocardial infarction | 112 (1.3%) | 66 (58.9%) | 60 (53.6%) | 69 (61.6%) | 70 (62.5%) |
| Deep vein thrombosis | 88 (1.0%) | 18 (20.5%) | 15 (17.0%) | 23 (26.1%) | 27 (30.7%) |
| Myocarditis | 38 (0.4%) | 13 (34.2%) | 11 (28.9%) | 14 (36.8%) | 14 (36.8%) |
| Lung edema | 15 (0.2%) | 2 (13.3%) | 0 (0.0%) | 2 (13.3%) | 3 (20.0%) |

Data are n (%), median (IQR) or mean (SD). BMI: Body Mass Index; ARDS: Acute respiratory distress syndrome.

## Factors associated with all-cause mortality

The results of the logistic regression model are presented for all covariates significantly associated with 180-day all-cause mortality in Fig 4. The model had good discrimination (AUC = 0.81; 95%-CI 0.80 to 0.82) and fit ($R^2$ = 0.23). Adjusted for age, sex and comorbidities, strong associations with increased odds of 180-day all-cause mortality (OR > 2) were observed for patients with a BMI $\geq$ 40 (OR 2.01, 95%-CI 1.33 to 3.05), liver disease (OR 2.45, 95%-CI 1.85 to 3.25), metastatic cancer (OR 8.02, 95%-CI 3.57 to 18.00), and coagulopathy (OR 2.31. 95%-CI 1.82 to 2.94). For age, the OR was 1.08 per year, indicating that the odds for 180-day all-cause mortality increase by the factor 2.21 per additional 10 years of age (1.082449^10), 4.88 per additional 20 years of age (1.082449^20), and so on. Conversely, a strong association for decreased odds of 180-day all-cause mortality was observed for female patients (OR 0.63, 95%-CI 0.56 to 0.70), and patients with depression (OR 0.46, 95%-CI 0.37 to 0.57).

## Readmissions

Of the 6,518 patients discharged alive from index hospitalization, 283 patients were lost to follow-up, as they died after 180 days of initial admission but before 180 days post discharge or were not insured with AOK anymore. Of the remaining 6,235 patients, 1,668 were readmitted a total of 2,551 times for some cause within180 days post discharge, resulting in an overall readmission rate of 26.8%.

Of the 6,518 patients discharged alive, 405 (6.2%) patients died within 180 days of the initial admission. Around half of them (201/405; 49.6%) were readmitted within 180 days of discharge, while the other half was not (204/405; 50.4%). Thus, the increase in 180-day all-cause mortality post discharge from the initial hospitalization was much higher in readmitted patients (201/1,668; 12.1%) than in those not readmitted (204/4,850; 4.2%).

Unlike mortality rates, readmission rates were only slightly higher among men (893/3,204; 27.9%) than women (775/3,031; 25.6%) (Table 2). Patients treated with IMV (231/717; 32.2%)

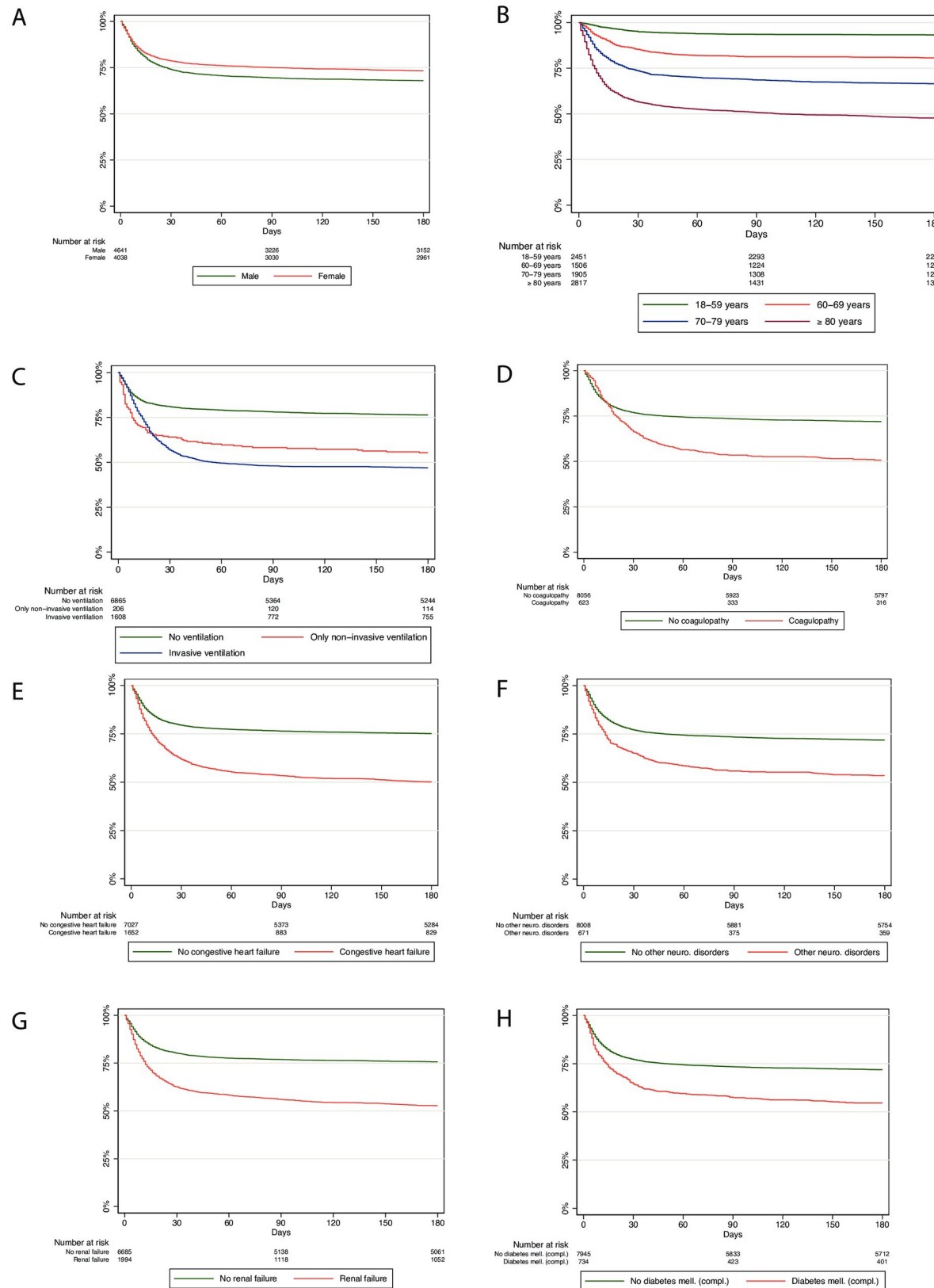

**Fig 2. Kaplan-Meier survival curves of all hospitalized COVID-19 patients followed for 180 days after hospital admission.**

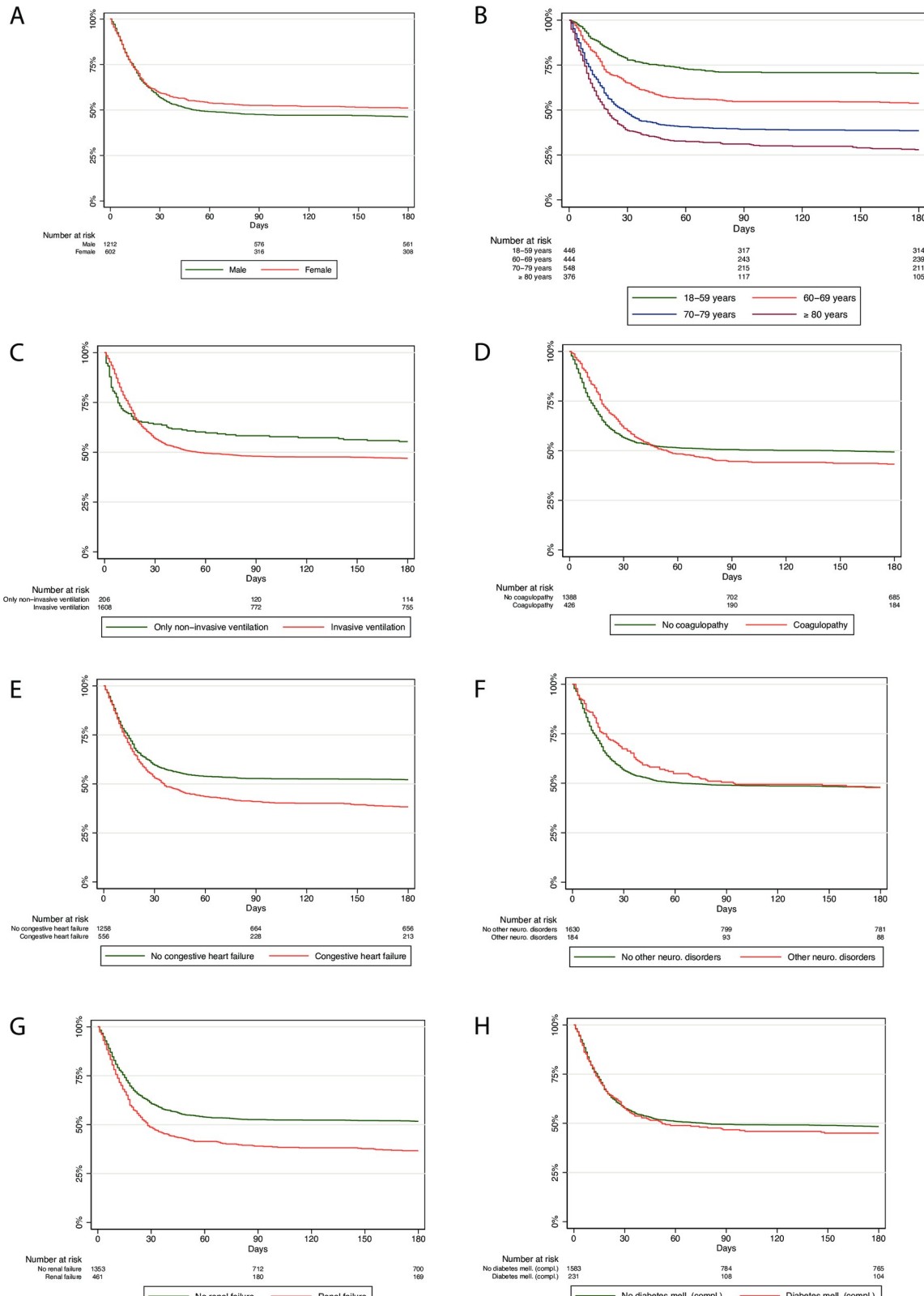

**Fig 3. Kaplan-Meier survival curves of all hospitalized COVID-19 patients on mechanical ventilation followed for 180 days after hospital admission.**

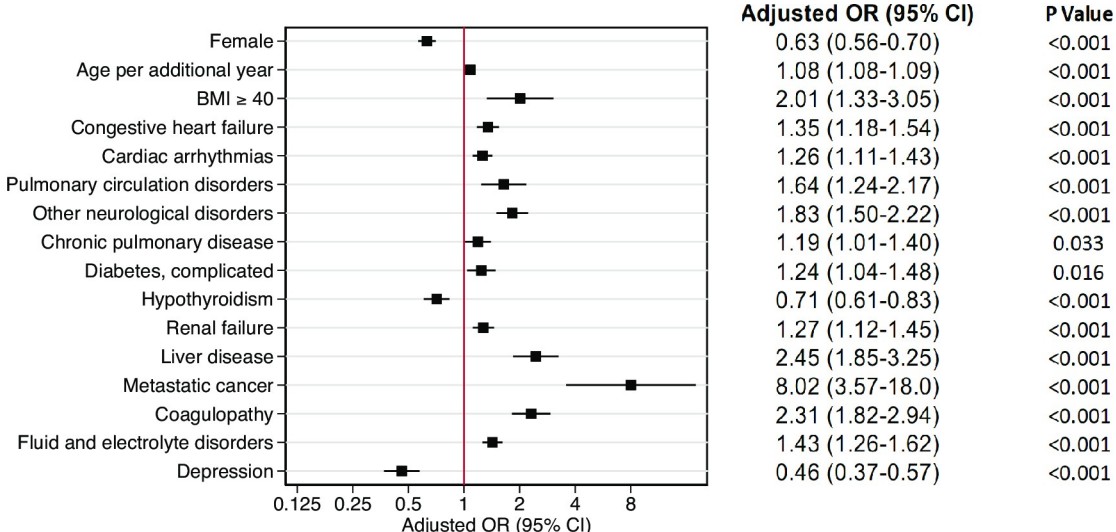

**Fig 4. Multivariable logistic regression analysis for 180-day all-cause mortality after hospital admission.** CI confidence interval, OR odds ratio, BMI body mass index. Only significant risk factors are included in the model. Each covariate has been adjusted for all other covariates displayed.

had higher readmission rates than patients treated with NIV (34/121; 28.1%) or patients who were not ventilated (1,403/5,397; 26.0%).

The majority of the readmitted patients (1,011/1,668; 60.6%), who had potentially COVID-19-related systemic, respiratory, renal, neurological, gastrointestinal liver, and cardio vascular principal/secondary diagnoses, were ventilated at readmission and/or had a positive PCR test

**Table 2. Readmission within 180-days of discharge.**

| Readmission among patients discharged alive from index admission | Total | Male | Female |
|---|---|---|---|
| | (n = 6,235) | (n = 3,204) | (n = 3,031) |
| All patients with at least one readmission | 1,668 (26.8%) | 893 (27.9%) | 775 (25.6%) |
| Patients with ventilation during index admission | | | |
| *Exclusively non-invasive ventilation* | | | |
| 34/121 (28.1%) | | 20/78 (25.6%) | 14/43 (32.6%) |
| *Invasive ventilation* | | | |
| 231/717 (32.2%) | | 154/453 (34.0%) | 77/264 (29.2%) |
| *No ventilation* | | | |
| 1,403/5,397 (26.0%) | | 719/2,673 (26.9%) | 684/2,724 (25.1%) |
| COVID-19-related readmission within 180 days | Principal or secondary diagnosis (% of readmitted patients) | | |
| Any systemic/ respiratory/ renal/ neuro/ gastrointestinal and liver/ cardiovascular/ ventilation disorder or complication/ COVID-19 | 1,011 (60.6%) | 550 (61.6%) | 461 (59.5%) |
| Systemic disorders/complications | 162 (9.7%) | 104 (11.6%) | 58 (7.5%) |
| Respiratory disorders/complications | 601 (36.0%) | 351 (39.3%) | 250 (32.3%) |
| Renal disorders/complications | 203 (12.2%) | 119 (13.3%) | 84 (10.8%) |
| Neurological disorders/complications | 490 (29.4%) | 244 (27.3%) | 246 (31.7%) |
| Gastrointestinal and liver disorders/complications | 117 (7.0%) | 77 (8.6%) | 40 (5.2%) |
| Cardiovascular disorders/complications | 192 (11.5%) | 115 (12.9%) | 77 (9.9%) |
| Mechanical ventilation (non-invasive and invasive) | 103 (6.2%) | 66 (7.4%) | 37 (4.8%) |
| COVID-19 (U07.1!) | 212 (12.7%) | 124 (13.9%) | 88 (11.4%) |

for COVID-19 (Table 2). When viewed separately, the most frequent principal or secondary diagnoses for readmission were respiratory disorders or complications (601/1,668; 36.0%), and neurological disorders or complications (490/1,668; 29.4%). Less than ten percent of the patients received mechanical ventilation at readmission (103/1,668; 6.2%). 212 (12.7%) patients tested again positive for COVID-19 at readmission.

## Discussion

This is the first study showing the 6-month all-cause mortality of hospitalized COVID-19 patients in a nationwide cohort, including readmissions within this time period. The major findings are the high 6-month all-cause mortality in COVID-19 patients, particularly among those requiring mechanical ventilation at 53%, that women have a sustained and profoundly beneficial 6-month outcome compared to men, and that patients aged above 80 years have the worst outcome with a 6-month all-cause mortality rate of 52% or up to 71% for those being ventilated. Furthermore, coagulopathy, liver diseases and severe obesity are associated with poor 6-month all-cause mortality. Lastly, readmission rates reached 27% with most of the patients with potentially COVID-19-related diagnoses being admitted for respiratory or neurological disorders or complications.

The initial in-hospital all-cause mortality of 25%, which is in line with many other European countries, increased to an all-cause mortality of 30% after 6 months, which demonstrates severe major prolonged implications of this disease, rather more than we would have expected. It is noticeable that ventilated patients have a poor overall outcome, especially patients over 70 years of age. In contrast, it is also evident that, in the younger age groups, there is only a slight increase in all-cause mortality after hospital discharge, although serious long-term consequences, having a huge impact on morbidity and quality of life may occur.

With regard to risk factors being associated with a poor or a more beneficial long-term outcome, several key factors became obvious. Overall, women show a better long-term outcome than men regardless of other confounding factors in terms of all-cause mortality, which may be due to their enhanced immune and inflammatory response to COVID-19 compared to men [23–26]. This is likely caused by their different genetic and endocrine mechanisms, including sex hormone actions, which might also influence the mechanisms of coagulopathy and thrombosis in COVID-19 [27]. On the other hand, factors contributing to increased all-cause mortality are especially disorders of the coagulation system, or liver disease, as already known from in-hospital mortality. These data are in line with our current understanding of COVID-19, particularly regarding the disorders of the coagulation or cerebrovascular system [28–30].

While diabetes generally worsens the outcome, this is not the case for patients being mechanically ventilated, whereas acute renal failure, congestive heart failure and age account for a worse outcome, independent of being ventilated. However, further data on patients' diagnoses before admission with COVID-19 are absent in this study. In the light of the current analysis, it should be critically evaluated whether current intensive care therapy, including mechanical ventilation in patients over 80 years of age, is really effective in view of the very high mortality rate, or whether, in future, we should develop narrow criteria for eligible patients so that they could have a more favorable outcome. This certainly includes the frailty of the elderly [31–33].

Within 180 days of discharge, there was a high number of readmissions, representing of 27% of those discharged alive, primarily due to respiratory or neurological diagnoses, independent of gender. Also, all-cause mortality in readmitted patients remains rather high at 6% of all discharged patients. Besides the lung, readmissions with neurological complications may be a

severe manifestation of the disease which may last for several months and lead to long-term complications, with their outcomes remaining unknown. Patients with coagulopathy had the highest 180-day all-cause mortality, which sheds a special light on the early detection of possible thrombosis or pulmonary embolism following the initial admission. In general, a close follow-up by general practitioners and the corresponding specialized disciplines is needed for early interventions regarding neurocognitive impairments. Notably, 13% of all readmitted patients were still or again positive for SARS-CoV-2, pointing out that virus elimination may take a long time in some severely ill patients.

## Strengths and limitations

The major strength of our study is its size and length of follow-up, our cohort of being the largest cohort of patients hospitalized for COVID-19 that was followed up for six months after admission and discharge, respectively. The data source (claims data) also has several strengths. First, due to the administrative nature of the data, we present an almost complete cohort of 98% of the eligible patients with full follow-up data. Furthermore, information on readmission is independent of whether readmission was handled by the same or a different hospital. Second, since the choice of hospital is free under German statutory health insurance, our data set includes hospitals ranging from major tertiary referral centers to smaller regional hospitals, comprising real-world data unbiased by the degree of hospital specialization. Third, in-patient data is of high quality because disease and procedure codes are relevant for the amount of reimbursement and are, therefore, verified by hospitals and sickness funds. Lastly, unlike most studies using administrative data, we were able to assign data from different hospital stays to the individual patient so that the unit of analysis was the patient and not the hospital case.

There are also several limitations relating to the data source. First, it only includes patients from one group of German sickness fund. However, it is the largest group, which accounts for one-third of the total population, providing a large sample representative of the German population. Second, patient-specific data are limited to in-patient diagnoses, procedures, and initial characteristics, so some pre-existing conditions might have remained unknown. Third, we stratified by mechanical ventilation, and not by ICU treatment (as it is not coded separately), which sometimes includes high-flow oxygen therapy without mechanical ventilation. Fourth, detailed data such as laboratory values and information on patient preferences or clinical decision-making that may impact the initiation of invasive treatments are not available.

General limitations include that, given its observational nature, this study cannot determine causality between risk factors and long-term mortality, nor does it provide information on the patients' cause of death. Lastly, the inclusion of patients admitted for COVID-19-related principal diagnoses might not be sufficient to distinguish between patients who were hospitalized for COVID-19 and patients with COVID-19 who were hospitalized for other reasons. Notably, 6-month all-cause mortality was similar between the in- and excluded patients.

## Conclusions

In this nationwide cohort of patients hospitalized for COVID-19, considerable long-term all-cause mortality and readmission rates were observed. Patients with coagulopathy had the highest increase in 180-day all-cause mortality, followed by congestive heart failure, neurological diseases, and acute renal failure. However, the female sex is a profoundly protective factor in the COVID-19 disease.

## Supporting information

**S1 Table. Diagnosis codes of included patients.**
(DOCX)

**S2 Table. Patient characteristics stratified by ventilation status.**
(DOCX)

## Author Contributions

**Conceptualization:** Christian Günster, Reinhard Busse, Gerhard Schillinger, Christian Karagiannidis.

**Data curation:** Christian Günster, Melissa Spoden, Tanja Rombey.

**Formal analysis:** Christian Günster, Reinhard Busse, Melissa Spoden, Gerhard Schillinger, Wolfgang Hoffmann, Steffen Weber-Carstens, Christian Karagiannidis.

**Methodology:** Christian Günster, Reinhard Busse, Melissa Spoden, Tanja Rombey, Gerhard Schillinger, Steffen Weber-Carstens, Andreas Schuppert, Christian Karagiannidis.

**Project administration:** Gerhard Schillinger, Christian Karagiannidis.

**Resources:** Reinhard Busse, Gerhard Schillinger.

**Supervision:** Reinhard Busse, Gerhard Schillinger, Wolfgang Hoffmann, Steffen Weber-Carstens, Andreas Schuppert, Christian Karagiannidis.

**Validation:** Christian Günster, Reinhard Busse, Melissa Spoden, Tanja Rombey, Gerhard Schillinger, Steffen Weber-Carstens, Andreas Schuppert, Christian Karagiannidis.

**Visualization:** Christian Günster, Melissa Spoden.

**Writing – original draft:** Christian Günster, Reinhard Busse, Tanja Rombey, Christian Karagiannidis.

**Writing – review & editing:** Wolfgang Hoffmann, Steffen Weber-Carstens, Andreas Schuppert.

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
