## [Decision Letter · Decision Letter 0]

15 Jun 2021

PONE-D-21-15090

6-month follow up of hospitalised COVID-19 patients: A nationwide cohort study of 8,679 patients in Germany

PLOS ONE

Dear Dr. Karagiannidis,

Thank you for submitting your manuscript to PLOS ONE. After careful consideration, we feel that it has merit but does not fully meet PLOS ONE’s publication criteria as it currently stands. Therefore, we invite you to submit a revised version of the manuscript that addresses the points raised during the review process.

We look forward to receiving your revised manuscript.

Kind regards,

Aleksandar R. Zivkovic

Academic Editor

PLOS ONE

"RB reports grants from Berlin University Alliance, during the conduct of the study;

grants from Federal Ministry of Research and Education, grants from Federal Ministry

of Health, grants from Innovation Fonds of the Federal Joint Committee, grants from

World Health Organization, outside the submitted work, AS reports grants from Bayer

AG, outside the submitted work. CK reports personal fees from Maquet, personal fees

from Xenios, personal fees from Bayer, non-financial support from Speaker of the

German register of ICUs, grants from German Ministry of Research and Education,

during the conduct of the study. CG, MS, TR, GS, WH, and SWC have nothing to

disclose."

4. Please include your tables as part of your main manuscript and remove the individual files. Please note that supplementary tables (should remain/ be uploaded) as separate "supporting information" files

Reviewers' comments:

Reviewer's Responses to Questions

Reviewer #1: I assessed the manuscript by Günster and co-workers which aimed to assess factors associated to 6 months outcomes after discharged in COVID-19 hospitalized patients starting from the dataset of German Local Health Fundus. The topic of medium and long term outcomes of hospitalized COVID-19 patients is for sure of interest considering the magnitude of the pandemic with a high number of patients requiring intensive care with potential long term detrimental effect on health status. The manuscript is not easy to read and does not follow the STROBE checklist for observational study with a very poor method section. The main problem of the manuscript is a study design and outcome definition/selection which are unable to answer the research question posed by the authors.

Major comments:

- The major outcome of interest is death. Nevertheless, this is not stated by the authors nor in the abstract nor at the end of the introduction section.

- It is surprising to read “unselected and unbiased cohort” at the end of the introduction section. In particular, it is clear that the major problem of the study is a selection bias introduced by the authors with the exclusion into the analysis of 2,780 of patients who “does not suffer from a COVID-19 principal diagnosis”

- The dataset is insufficient to answer the research question. In particular, the authors were unable to distinguish between COVID-19 related and unrelated deaths.

- The definition of the effect modified entered into the model is poor. Several factors seem to be pre-hospital determinants of COVID-19 worse outcomes (i.e. age, gender, BMI) whereas other (i.e. mechanical ventilation, shock, etc) are events which happened during the first index events. Nevertheless, no mention across the manuscript was made about how time-to-events was handled into the final mode.

- It is surprising to see different median time of hospital stay in Table 1 for the “index hospital admission” at different time point (30-90-180 days). How is it possible if the baseline study population is the same?

Reviewer #2: The study includes a large amount of COVID-19 patients admitted to the hospital in Germany and analyses risk factors for in-hospital mortality and post-discharge mortality, as well as for readmission. This study addresses an important and current issue which goes beyond acute disease. Specifically, it shows a relevant impact of COVID-19 even after acute disease resolution and points to the importance of post-discharge patient monitoring to precede long-term mortality and readmission rates.

The large sample size is certainly an important advantage of this work. However, the limited length of stay needed for patient inclusion (February-April 2020) may create selection bias by excluding those patients with longer length of stay, who may be those with more severe disease or transferred to the intensive care unit.

There are few major issues to address:

- In the manuscript there a lot of English errors. Therefore, I suggest the manuscript to be thoroughly reviewed by a native English-speaker before resubmission.

- It is unclear why Cox regression was not used to build the predictive models and logistic regression was used instead. I would suggest to implement the methods section to better specify this. Time is an important factor to consider and a time-dependent analysis would be preferred, although logistic regression may still be fine since several time points are considered.

- It would be interesting to test the model in an external cohort. Alternatively, a method of internal validation should be considered to confirm the results.

- In the depicted Kaplan Meier curves in figure 2.

- In figure 3, it is unclear which are the variables which covariates are adjusted for.

Reviewer #3: In this nationwide observational study, the authors provide a detailed account of the follow-up of hospitalized COVID-19 patients until 6-months after their initial hospital admission. There is increasing information now in the literature about the outcome of hospitalized COVID-19 patients after discharge. However, the novelty of the present study rests on the large cohort considered with 6-month follow-up. Indeed, with the exception of China, so far published data on large cohorts of hospitalized COVID-19 patients limited their observation up to 3 months.

There are few areas, which are of concern for the authors considerations:

1. Here the focus is on mortality during 6-month follow-up, including death occurring during possible readmission to hospital after discharge. However, they did not analyze others outcomes such as long-term complications and incomplete recovery after hospital discharge. Therefore, the title of the manuscript should be more specific, including the reference to the parameter ‘mortality’ they have analyzed.

2. Data were from the German local health care funds, a health insurance system that approximately cover 32% of the German population. Although briefly mentioned in the limitations of the study, it would be useful to provide more details on the representativeness of the study cohort for all the German population, given the term ‘nationwide’ attributed to the study in the title, and before concluding for the generalization of the findings.

3. It would have been interesting to analyze separately the group of patients who were hospitalized for COVID-19 and that of COVID-19 patients hospitalized for other reasons. The authors, however, cannot distinguish these two groups. Therefore, this issue should be discussed in the limitations’ paragraph.

4. As shown in figure 2d, the 30-day mortality for patients with coagulopathy was lower than that of patients without coagulopathy. How the authors may explain this unexpected finding?

5. In the Discussion, they should elaborate more on the reason why female sex is a protective factor in terms of mortality in COVID-19 disease, as documented in the present study.

6. In the Discussion section, they need to highlight the strength and novelty of the study, as well as to elaborate more on the limitations as indicated above.

Minor

1. In the first paragraph of the Introduction, the authors reported the rates of hospitalization and mortality in France, Spain, UK and Germany. In support, they quoted reference #4 and #5 among others, which however refer to Italy. This discrepancy should be fixed.

2. For each panel of Figure 1 and 2, the Kaplan-Meier survival curves should include index number of patients at each time points.

Reviewer #4: The authors present a retrospective observational study in Germany on 6-months mortality rate and outcomes of patients with hospitalised Covid-19. The data is clearly presented.

Although retrospective, the study includes a high number of patients, representative of the german population and whose characteristics (age, male predominance) are in accordance with other published series on Covd-19 epidemiology. The authors find that 6 months mortality is high, higher in men than women.

One limit of the study is that it included patients during the first epidemic wave, when mortality rate was probably higher than now, as stated by several studies, especially for patients with coagulopathy. Still, this study remains of interest as it describes the course of severe Covid-19 and since mortality remains quite high in hospitalised patients and as it includes a representative population. The evolution of mortality over time could be better emphasised in the discussion section.

One might question the high readmission rate, especially for neurological and respiratory conditions, since this does not seem to be the case in all countries : authors should discuss this point in light of discharge conditions in Germany (e.g. are patients discharged home or do they benefit from in-hospital readmission with a transfer to another hospital ?) and in light of their personnal experience of the causes of readmission (what stands under neurological and respiratory conditions ?)

6. PLOS authors have the option to publish the peer review history of their article (what does this mean?). If published, this will include your full peer review and any attached files.

Reviewer #1: No

Reviewer #2: No

Reviewer #3: No

Reviewer #4: **Yes: **Justine Frija-Masson

---

## [Author Response · Author response to Decision Letter 0]

13 Jul 2021

PLOS one

Editor in Chief

Prof. Zivkovic

Editor’s comments:

AUTHORS RESPONSE: Thank you for the opportunity to revise and resubmit our manuscript! We have formatted our manuscript according to the style requirements and renamed our files accordingly.

CHANGES TO THE MANUSCRIPT: To enhance the readability of our marked manuscript we did not track formatting changes. No formatting change had an influence on the written content.

AUTHORS RESPONSE: Unfortunately, there are legal restrictions on sharing the de-identified data set. We addressed this issue in our revised cover letter:

The data used in this study cannot be made available in the manuscript, the supplemental files, or in a public repository due to German data protection laws (Bundesdatenschutzgesetz). Therefore, they are stored on a secure drive in the Wissenschaftliches Institut der AOK to facilitate replication of the results. Generally, access to data of statutory health insurance funds for research purposes is possible only under the conditions defined in German Social Law (SGB V § 287). Requests for data access can be sent as a formal proposal specifying the recipient and purpose of the data transfer to the appropriate data protection agency. Access to the data used in this study can only be provided to external parties under the conditions of the cooperation contract of this research project and after written approval by the AOK. For assistance in obtaining access to the data, please contact wido@wido.bv.aok.de.

CHANGES TO THE MANUSCRIPT: We have added a data availability statement containing the information provided above (ll. X-xx).

"RB reports grants from Berlin University Alliance, during the conduct of the study; grants from Federal Ministry of Research and Education, grants from Federal Ministry of Health, grants from Innovation Fonds of the Federal Joint Committee, grants from World Health Organization, outside the submitted work, AS reports grants from Bayer AG, outside the submitted work. CK reports personal fees from Maquet, personal fees from Xenios, personal fees from Bayer, non-financial support from Speaker of the German register of ICUs, grants from German Ministry of Research and Education, during the conduct of the study. CG, MS, TR, GS, WH, and SWC have nothing to disclose."

AUTHORS RESPONSE: Thank you for pointing this out. We have added the suggested statement in the revised cover letter and manuscript. We also added a data availability statement (see our response to your previous comment).

CHANGES TO THE MANUSCRIPT: Added “This does not alter our adherence to PLOS ONE policies on sharing data and materials.” (ll. X-xx).

AUTHORS RESPONSE: Thank you. We confirm that we have understood the policy. We have included our updated Competing Interests statement.

CHANGES TO THE MANUSCRIPT: Changed according to the suggestions.

4. Please include your tables as part of your main manuscript and remove the individual files. Please note that supplementary tables (should remain/ be uploaded) as separate "supporting information" files

AUTHORS RESPONSE: Thank you, we followed your comment.

CHANGES TO THE MANUSCRIPT: Tables 1 and 2 are now included in the manuscript. Table S1 and S2 have been uploaded as a separate file.

AUTHORS RESPONSE: We have added captions for our Supporting Information files at the end of our manuscript and have updated in-text citations accordingly.

CHANGES TO THE MANUSCRIPT: Added “Supporting Information: Table S1: Diagnosis codes of included patients. Table S2: Patient characteristics stratified by ventilation status.” (ll. X-xx)

Reviewers' comments:

Reviewer's Responses to Questions

Reviewer #1: I assessed the manuscript by Günster and co-workers which aimed to assess factors associated to 6 months outcomes after discharged in COVID-19 hospitalized patients starting from the dataset of German Local Health Fundus. The topic of medium and long term outcomes of hospitalized COVID-19 patients is for sure of interest considering the magnitude of the pandemic with a high number of patients requiring intensive care with potential long term detrimental effect on health status. The manuscript is not easy to read and does not follow the STROBE checklist for observational study with a very poor method section. The main problem of the manuscript is a study design and outcome definition/selection which are unable to answer the research question posed by the authors.

AUTHORS RESPONSE: Thank you very much for reviewing our manuscript! We have re-organized our manuscript to follow the STROBE checklist more closely. We particularly restructured and supplemented the methods section.

CHANGES TO THE MANUSCRIPT: We inserted additional subject headings throughout the manuscript, rearranged the methods section, and added the following sentences:

ll. X-x: “This was a retrospective observational cohort study using claims data.”

ll. X-x: “Bias: Thorough legal requirements and verification from both hospitals and sickness funds aim to minimize bias introduced by false coding and other systematic errors relating to the data. In addition, we checked data plausibility and consistency before analyzing the data.”

ll. x-x: “Study size: This was a complete survey of all AOK insured.”

ll. x-x: “Due to non-parallel Kaplan-Meier curves and a significant deviation from the proportional hazards assumption, no Cox-proportional hazard model could be estimated. The proportional hazards assumption was evaluated visually on plots of log(-log(survival)) versus log of survival time adjusted for covariates and by a global test based on Schoenfeld residuals. Instead, […].”

ll. x-x: “We used cluster-robust standard errors in order to account for clustering of patients in hospitals. Comorbidity conditions were defined as binary variables.”

ll. x-x: “[…] and summarized in a forest plot. To evaluate the performance of the model, the area under the curve (AUC) was used as measure of discrimination and the squared Pearson correlation (R2) between 180-day all-cause mortality and the log-odds of predicted mortality was used as measure of explained variation.”

Major comments:

- The major outcome of interest is death. Nevertheless, this is not stated by the authors nor in the abstract nor at the end of the introduction section.

AUTHORS RESPONSE: Thank you for pointing this out! We have now added this information in the abstract and end of introduction section. Furthermore, we changed the title to “6-month mortality and readmissions of hospitalised COVID-19 patients: a nationwide cohort study of 8,679 patients in Germany” to better reflect the focus of our study.

CHANGES TO THE MANUSCRIPT: 

ll. 4-6 (Title): “6-month mortality and readmissions of hospitalised COVID-19 patients: a nationwide cohort study of 8,679 patients in Germany”

ll. X-x (Abstract, Methods): “…., for whom 6-month all-cause mortality and readmission rates for 180 days after admission or until death was available.” 

ll. Xx-x (Introduction, Objectives): “[…], the aim of this observational study was to determine 6-month all-cause mortality and readmission rates of hospitalised COVID-19 patients with completed hospital treatments and a confirmed COVID-19 diagnosis, […] Furthermore factors associated with 6-month all-cause mortality were evaluated.”

- It is surprising to read “unselected and unbiased cohort” at the end of the introduction section. In particular, it is clear that the major problem of the study is a selection bias introduced by the authors with the exclusion into the analysis of 2,780 of patients who “does not suffer from a COVID-19 principal diagnosis”

AUTHORS RESPONSE: We agree with the reviewer’s point and have revised the objectives statement accordingly. Furthermore, we now explain and justify our selection of patients in the methods section and discuss the issue in the limitations section of our revised manuscript.

CHANGES TO THE MANUSCRIPT: 

Deleted: “in a large, unselected and unbiased cohort of patients with”

Added: 

ll. Xx-x (Methods, Participants): “The selection of patients was performed to include only patients in whom COVID-19 was the primary reason for their hospital stay and to exclude those in whom COVID-19 was an incidental finding likely to be unrelated to their hospital stay.”

ll. x-x (Discussion, Strength and limitations): “Lastly, the inclusion of patients admitted for COVID-19-related principal diagnoses might not be sufficient to distinguish between patients who were hospitalised for COVID-19 and those COVID-19 patients hospitalised for other reasons.”

- The dataset is insufficient to answer the research question. In particular, the authors were unable to distinguish between COVID-19 related and unrelated deaths.

AUTHORS RESPONSE: While we agree with the reviewer that it is generally very important distinguish between COVID-19-related and -unrelated deaths, we would like to stress that it was not our aim to assess the patients’ cause of death but to assess all-cause mortality in patients hospitalised for COVID-19. As already stated in the limitations section, our study was purely observational and does not allow for establishing causality. We have added a subclause to clarify this issue. Furthermore, we now consistently use the term “all-cause mortality” throughout the manuscript.

CHANGES TO THE MANUSCRIPT: Added: Ll. Xx.x: “[…] this study cannot determine causality between risk factors and long-term mortality, nor does it provide information on the patients’ cause of death.”

- The definition of the effect modified entered into the model is poor. Several factors seem to be pre-hospital determinants of COVID-19 worse outcomes (i.e. age, gender, BMI) whereas other (i.e. mechanical ventilation, shock, etc) are events which happened during the first index events. Nevertheless, no mention across the manuscript was made about how time-to-events was handled into the final mode.

AUTHORS RESPONSE: Thank you for your thorough review, which has called our attention to interconnected weak spots in our presentation of methods and results. Patient characteristics and strata were solely defined by conditions and interventions during index hospitalization. Thereafter comorbidities at index hospitalization were entered into the multivariable modelling of 6-month all-cause mortality. Apart from the Kaplan-Meier analysis, we did use multivariable modelling for the binary endpoint of 180-day all-cause mortality after admission. Cox regression could not be applied to handle time-to-event more detailed because of a major deviation from the proportional hazards assumption. Therefore, we estimated a model by logistic regression. We are grateful for having had this pointed out and have sought to improve both sections in terms of clarity.

CHANGES TO THE MANUSCRIPT: We rearranged the methods section. We added

ll. Xx.x “Baseline characteristics are defined on conditions and interventions during index hospitalization […].”

ll. x-x: “Due to non-parallel Kaplan-Meier curves and a significant deviation from the proportional hazards assumption, no Cox-proportional hazard model could be estimated. The proportional hazards assumption was evaluated visually on plots of log(-log(survival)) versus log of survival time adjusted for covariates and by a global test based on Schoenfeld residuals.”

We updated ll x-x: “Instead, multivariable logistic regression was used to model the odds of the binary endpoint 180-day all-cause mortality after admission as function of age, sex, body mass index (BMI) categories (30≤34, 35≤39, ≥40 kg/m2) and Elixhauser comorbidities present at index hospitalization.”

- It is surprising to see different median time of hospital stay in Table 1 for the “index hospital admission” at different time point (30-90-180 days). How is it possible if the baseline study population is the same?

AUTHORS RESPONSE: Thank you for pointing this out! The values refer to the respective column total. We have added the missing row descriptions.

CHANGES TO THE MANUSCRIPT: Table 1: “Age, years: (value refers to column total)”, “Length of stay of index hospitalisation: (value refers to column total)"

Reviewer #2: The study includes a large amount of COVID-19 patients admitted to the hospital in Germany and analyses risk factors for in-hospital mortality and post-discharge mortality, as well as for readmission. This study addresses an important and current issue which goes beyond acute disease. Specifically, it shows a relevant impact of COVID-19 even after acute disease resolution and points to the importance of post-discharge patient monitoring to precede long-term mortality and readmission rates.

The large sample size is certainly an important advantage of this work. However, the limited length of stay needed for patient inclusion (February-April 2020) may create selection bias by excluding those patients with longer length of stay, who may be those with more severe disease or transferred to the intensive care unit.

AUTHORS RESPONSE: In our study, we included patients admitted to hospital between February and April 2020 with a length of stay not exceeding 30 June 2020. Accordingly, we were able to include patients with a hospital stay up to 2 months (when enrolled 30 April) to up to 5 months (when enrolled 1 February 2020). 

The reason for the cut-off date being 30 June 2020 is that our analyses were performed in January 2021. Thus, we were able to include data up to 31 December 2020, which meant that – for a complete 6-month follow-up after discharge – we could include only patients discharged until 30 June 2020.

CHANGES TO THE MANUSCRIPT: Not applicable.

There are few major issues to address:

- In the manuscript there a lot of English errors. Therefore, I suggest the manuscript to be thoroughly reviewed by a native English-speaker before resubmission.

AUTHORS RESPONSE: Thank you for your suggestion. Our manuscript has now been reviewed by a professional company to correct any mistakes and improve its readability. Furthermore, the manuscript was adapted to American English. 

CHANGES TO THE MANUSCRIPT: There have been various corrections/changes to improve language throughout the manuscript (all of which are tracked in the marked version of the manuscript and none of which had an influence on the written content). 

Added: ll. X-x (Acknowledgements): The authors would like to thank NN for proof-reading and improving the language of our manuscript.

- It is unclear why Cox regression was not used to build the predictive models and logistic regression was used instead. I would suggest to implement the methods section to better specify this. Time is an important factor to consider and a time-dependent analysis would be preferred, although logistic regression may still be fine since several time points are considered.

AUTHORS RESPONSE: We thank you for pointing this out. We would have preferred to conduct a Cox regression to handle time-to-event information more detailed. But unfortunately, Cox regression could not be applied due to a major deviation from the proportional hazards assumption. Therefore, we estimated a model for the binary endpoint of 180-day all-cause mortality after admission by logistic regression. Models for the other time points (in hospital-/30-day/90-day- all-cause mortality) were estimated too and showed similar results with more pronounced impact of metastatic cancer for longer follow up (not shown).

CHANGES TO THE MANUSCRIPT: We added ll. x-x: “Due to non-parallel Kaplan-Meier curves and a significant deviation from the proportional hazards assumption, no Cox-proportional hazard model could be estimated. The proportional hazards assumption was evaluated visually on plots of log(-log(survival)) versus log of survival time adjusted for covariates and by a global test based on Schoenfeld residuals.”

We updated ll x-x: “Instead, multivariable logistic regression was used to model the odds of the binary endpoint 180-day all-cause mortality after admission as function of age, sex, body mass index (BMI) categories (30≤34, 35≤39, ≥40 kg/m2) and Elixhauser comorbidities present at index hospitalization.”

- It would be interesting to test the model in an external cohort. Alternatively, a method of internal validation should be considered to confirm the results.

- In the depicted Kaplan Meier curves in figure 2.

AUTHORS RESPONSE: Thank you for this interesting point. We wish to clarify that it was not our aim to develop a predictive model, but a multivariable logistic regression model of independent risk factors for 180-day all-cause mortality. We further specified the intent for and methods applying to our model in the abstract and methods section.

However, we are indeed planning compare this cohort to other cohorts, namely patients hospitalised during the “second wave” and “third wave”. It will be very interesting to see if the same independent risk factors for 180-day all-cause mortality will apply to these cohorts. Unfortunately, we are not able to perform these analyses as of now given the time frame until the respective data becomes available (see our response to your first comment). 

CHANGES TO THE MANUSCRIPT:

ll. x-x (Abstract): “A multivariable logistic regression model identified independent risk factors for 180-day all-cause mortality in this cohort.”

- In figure 3, it is unclear which are the variables which covariates are adjusted for.

AUTHORS RESPONSE: Thank you for pointing us to this ambiguity. Each covariate has been adjusted for all other covariates displayed in figure 3. We have added this information in the figure legend.

CHANGES TO THE MANUSCRIPT: Added: Figure legend Fig 3: “Each covariate has been adjusted for all other covariates displayed.”

Reviewer #3: In this nationwide observational study, the authors provide a detailed account of the follow-up of hospitalized COVID-19 patients until 6-months after their initial hospital admission. There is increasing information now in the literature about the outcome of hospitalized COVID-19 patients after discharge. However, the novelty of the present study rests on the large cohort considered with 6-month follow-up. Indeed, with the exception of China, so far published data on large cohorts of hospitalized COVID-19 patients limited their observation up to 3 months.

There are few areas, which are of concern for the authors considerations:

1. Here the focus is on mortality during 6-month follow-up, including death occurring during possible readmission to hospital after discharge. However, they did not analyze others outcomes such as long-term complications and incomplete recovery after hospital discharge. Therefore, the title of the manuscript should be more specific, including the reference to the parameter ‘mortality’ they have analyzed.

AUTHORS RESPONSE: Thank you very much for reviewing our manuscript! We agree with the reviewer and have changed the title accordingly.

CHANGES TO THE MANUSCRIPT: Title: “6-month mortality and readmissions of hospitalised COVID-19 patients: a nationwide cohort study of 8,679 patients in Germany”

2. Data were from the German local health care funds, a health insurance system that approximately cover 32% of the German population. Although briefly mentioned in the limitations of the study, it would be useful to provide more details on the representativeness of the study cohort for all the German population, given the term ‘nationwide’ attributed to the study in the title, and before concluding for the generalization of the findings.

AUTHORS RESPONSE: We followed the reviewer’s suggestion and have added more details on the German statutory health insurance system and the representativeness of the study cohort in the methods section.

CHANGES TO THE MANUSCRIPT: 

ll. Xx-x: “AOK is the largest sickness fund group within Germany’s statutory health insurance system. It provides statutory health insurance for roughly 32 percent of the German population, for whom it is representative when considering the factor age [20, 21]. Furthermore, membership in a sickness fund is open to anyone regardless of factors such as professional affiliation, income, or comorbidities. At the same time, sickness funds are obliged to accept any new member and charge the same basic contribution rate [19].”

3. It would have been interesting to analyze separately the group of patients who were hospitalized for COVID-19 and that of COVID-19 patients hospitalized for other reasons. The authors, however, cannot distinguish these two groups. Therefore, this issue should be discussed in the limitations’ paragraph.

AUTHORS RESPONSE: Thank you for your valuable suggestion. We agree that this kind of analysis would have been very interesting and is certainly a point that future research should address. We now discuss the issue in the limitations section of our manuscript.

CHANGES TO THE MANUSCRIPT: 

Added: ll. Xx-x: “Lastly, the inclusion of patients admitted for COVID-19-related principal diagnoses might not be sufficient to distinguish between patients who were hospitalized for COVID-19 and patients with COVID-19 who were hospitalized for other reasons. Of note, 6-months all-cause mortality was similar between the in- and excluded patients.”

4. As shown in figure 2d, the 30-day mortality for patients with coagulopathy was lower than that of patients without coagulopathy. How the authors may explain this unexpected finding?

AUTHORS RESPONSE: Thank you for raising this interesting question. We also wondered about the survival curves. However, our data don’t show a sufficient explanation. It might be that patient with recognized coagulopathy such as pulmonary embolism are under closer supervision within the first days of treatment. Therefore, initial survival might be somewhat better. Since this remains speculative, we didn’t integrate this point into the manuscript.

CHANGES TO THE MANUSCRIPT: Not applicable.

5. In the Discussion, they should elaborate more on the reason why female sex is a protective factor in terms of mortality in COVID-19 disease, as documented in the present study.

AUTHORS RESPONSE: We agree with the reviewer and have added another sentence and reference to discuss this important matter. 

CHANGES TO THE MANUSCRIPT: 

ll. X-x (Discussion): “Overall, women show a better long-term outcome than men regardless of other confounding factors in terms of all-cause mortality, which may be due to their enhanced immune and inflammatory response to COVID-19 compared to men [24-27]. This is likely caused by their different genetic and endocrine mechanisms, including sex hormone actions, which might also influence the mechanisms of coagulopathy and thrombosis in COVID-19 [28].”

6. In the Discussion section, they need to highlight the strength and novelty of the study, as well as to elaborate more on the limitations as indicated above.

AUTHORS RESPONSE: Thank you! We re-named the limitations section to “strengths and limitations” and have elaborated on both while considering your previous points.

CHANGES TO THE MANUSCRIPT: 

Added: ll.x-x (Discussion, Strengths and limitations): “The major strength of our study is its size and length of follow-up, our cohort of being the largest cohort of patients hospitalized for COVID-19 that was followed up for six months after admission and discharge, respectively. The data source (claims data) also has several strengths. First, due to the administrative nature of the data, we describe an almost complete cohort of 98% of the eligible patients with full follow up data. Furthermore, information on readmission is independent of whether readmission was handled by the same or a different hospital. Second, since choice of hospital is free under German statutory health insurance, our data set includes hospitals ranging from major tertiary referral centers to smaller regional hospitals, which affords real-world data unbiased by the degree of hospital specialization. Third, inpatient data is of high quality because disease and procedure codes are relevant for the amount of remuneration and are therefore verified by hospitals and sickness funds. Lastly, unlike most studies using administrative data, we were able to assign data from different hospital stays to the individual patient so that the unit of analysis was the patient not the hospital case.

There are also several limitations relating to the data source. […] General limitations include that, given its observational nature, this study cannot determine causality between risk factors and long-term mortality, nor does it provide information on the patients’ cause of death. Lastly, the inclusion of patients admitted for COVID-19-related principal diagnoses might not be sufficient to distinguish between patients who were hospitalized for COVID-19 and patients with COVID-19 who were hospitalized for other reasons. Of note, 6-months all-cause mortality was similar between the in- and excluded patients.”

Minor

1. In the first paragraph of the Introduction, the authors reported the rates of hospitalization and mortality in France, Spain, UK and Germany. In support, they quoted reference #4 and #5 among others, which however refer to Italy. This discrepancy should be fixed. 

AUTHORS RESPONSE: Thank you very much for pointing this out! We have corrected the respective references.

CHANGES TO THE MANUSCRIPT:

ll. x-x (References): 

“1. Department Health Care Management. Data on Covid-19 hospitalisations / ICU treatments across European countries with data available as of 17/03/21 Berlin: Technische Universität Berlin; 2021. Available from: https://www.mig.tu-berlin.de/fileadmin/a38331600/sonstiges/COVID-19-STATS_170321_1945.pdf. Last accessed: 27 June 2021.

2. Bonnet G, Weizman O, Trimaille A, Pommier T, Cellier J, Geneste L, et al. Characteristics and outcomes of patients hospitalized for COVID-19 in France: The Critical COVID-19 France (CCF) study. Arch Cardiovasc Dis. 2021. doi: 10.1016/j.acvd.2021.01.003.

3. Karagiannidis C, Mostert C, Hentschker C, Voshaar T, Malzahn J, Schillinger G, et al. Case characteristics, resource use, and outcomes of 10 021 patients with COVID-19 admitted to 920 German hospitals: an observational study. Lancet Respir Med. 2020;8(9):853-62. doi: 10.1016/S2213-2600(20)30316-7.

4. Docherty AB, Harrison EM, Green CA, Hardwick HE, Pius R, Norman L, et al. Features of 20 133 UK patients in hospital with covid-19 using the ISARIC WHO Clinical Characterisation Protocol: prospective observational cohort study. Bmj. 2020;369:m1985. doi: 10.1136/bmj.m1985.”

2. For each panel of Figure 1 and 2, the Kaplan-Meier survival curves should include index number of patients at each time points.

AUTHORS RESPONSE: Thank you for your suggestion. We have now added the number of patients at start, 90 and 180 days follow-up.

CHANGES TO THE MANUSCRIPT: Figure 1 and 2 (now 2 and 3): Changed accordingly.

Reviewer #4: The authors present a retrospective observational study in Germany on 6-months mortality rate and outcomes of patients with hospitalised Covid-19. The data is clearly presented.

Although retrospective, the study includes a high number of patients, representative of the german population and whose characteristics (age, male predominance) are in accordance with other published series on Covd-19 epidemiology. The authors find that 6 months mortality is high, higher in men than women.

One limit of the study is that it included patients during the first epidemic wave, when mortality rate was probably higher than now, as stated by several studies, especially for patients with coagulopathy. Still, this study remains of interest as it describes the course of severe Covid-19 and since mortality remains quite high in hospitalised patients and as it includes a representative population. The evolution of mortality over time could be better emphasised in the discussion section.

AUTHORS RESPONSE: Thank you very much for reviewing our manuscript!

The reviewer raised an important point. We are indeed planning to compare this cohort to patients hospitalised during the “second wave” and “third wave”. However, the follow-up period of our study being 6 months, we are not able to analyze patients from the second wave (which in Germany lasted until mid Feburary 2021) before autumn. Therefore, we chose not to discuss this issue in our current manuscript. 

However, we agree with the reviewer that it will be very interesting to see if mortality rates did in fact become lower in these subsequent waves. On the one hand, there is evidence that in-hospital mortality rates became lower. On the other hand, the later pandemic waves brought new variants of the virus with unknown long-term consequences. Furthermore, we found that there was no difference in in-hospital mortality rates between the first and second wave of ICU patients in Germany who we analysed as part of another study (Karagiannidis C, Windisch W, McAuley DF, Welte T, Busse R. Major differences in ICU admissions during the first and second COVID-19 wave in Germany. Lancet Respir Med. 2021;9(5):e47-e48. doi:10.1016/S2213-2600(21)00101-6).

CHANGES TO THE MANUSCRIPT: Not applicable.

One might question the high readmission rate, especially for neurological and respiratory conditions, since this does not seem to be the case in all countries: authors should discuss this point in light of discharge conditions in Germany (e.g. are patients discharged home or do they benefit from in-hospital readmission with a transfer to another hospital?) and in light of their personal experience of the causes of readmission (what stands under neurological and respiratory conditions ?)

AUTHORS RESPONSE: Thank you for this important comment. Compared to different health care systems, the German readmission rate is almost in average with other systems and with other diseases within our health care system with a broad variation. Since this is an everyday experience and we have only few data, we cannot add this to the manuscript, but we now added a new table with readmission reasons to the supplement.

CHANGES TO THE MANUSCRIPT: We now added a supplement table with definitions of all readmission groups.

---

## [Editor Report · Decision Letter 1]

16 Jul 2021

6-month mortality and readmissions of hospitalized COVID-19 patients: a nationwide cohort study of 8,679 patients in Germany

PONE-D-21-15090R1

Dear Dr. Karagiannidis,

We’re pleased to inform you that your manuscript has been judged scientifically suitable for publication and will be formally accepted for publication once it meets all outstanding technical requirements.

Kind regards,

Aleksandar R. Zivkovic

Academic Editor

PLOS ONE

---

## [Editor Report · Acceptance letter]

22 Jul 2021

PONE-D-21-15090R1 

6-month mortality and readmissions of hospitalized COVID-19 patients: a nationwide cohort study of 8,679 patients in Germany 

Dear Dr. Karagiannidis:

I'm pleased to inform you that your manuscript has been deemed suitable for publication in PLOS ONE. Congratulations! Your manuscript is now with our production department. 

Kind regards, 

on behalf of

Dr. Aleksandar R. Zivkovic 

Academic Editor

PLOS ONE